# Spring 2021 sea ice transport in the southern Beaufort Sea occurred during coastal lead opening events

MacKenzie E. Jewell[1], Jennifer K. Hutchings[2], and Angela C. Bliss[3]

[1,2]College of Earth, Ocean, and Atmospheric Sciences, Oregon State University, Corvallis, OR, USA
[3]NASA Goddard Space Flight Center, Cryospheric Sciences Laboratory, Greenbelt, MD, USA

**Correspondence:** MacKenzie E. Jewell (jewellm@oregonstate.edu)

**Abstract.** Winds blew a record large portion of the Arctic's multiyear sea ice (MYI) into the southern Beaufort Sea (SBS) in winter 2021. From early March, a network of buoys from the Sea Ice Dynamic Experiment (SIDEx) tracked the MYI as it drifted across the SBS toward the Chukchi Sea. Transport was episodic as the consolidated ice pack interacted with coastal boundaries and repeatedly fractured. We investigated variability in 2021 MYI transport by relating in situ sea ice drift to remotely sensed coastal lead opening events, which have been associated with increased ice drift speeds in winter. Daily ice concentration data show ten opening events occurred throughout March and April 2021. Opening lasted 1-5 consecutive days as southeasterly winds pushed the SBS ice pack away from coastal boundaries. During opening, the ice pack abruptly accelerated and its response to wind forcing stabilized around free drift conditions, drifting at $2.1\%$ of wind speeds (median rate of $13.7\,\mathrm{km\,d^{-1}}$). With this efficient wind-to-ice momentum transfer, nearly all ($94\%$) alongshore MYI transport in March-April 2021 occurred during opening events, which comprised just half of March-April days. Only $6\%$ of transport occurred during the other half of days without observed opening. On these days, the ice was mostly stationary (median $1.7\,\mathrm{km\,d^{-1}}$ drift rate, $0.6\%$ of wind speeds) as northwesterly winds compressed the pack against the coast. Spatiotemporal discontinuities in ice-wind speed ratios were measured across several coastal leads that transected the SIDEx buoy network, providing direct observations of changes in ice drift regime as coastal leads opened. These results quantify the disproportionate contribution of offshore and alongshore winds in breaking the SBS ice pack away from the coast, opening leads, and driving SBS ice transport during spring 2021, highlighting the critical role these episodic events play in consolidation season sea ice drift.

## 1 Introduction

In late February 2021, a record large portion of the Arctic Ocean's perennial sea ice cover resided in the Beaufort Sea (Mallett et al., 2021). In the preceding winter months, winds of record anticyclonic vorticity had blown over the Arctic sea ice cover (Mallett et al., 2021). These winds accelerated ice circulation around the southern leg of the Beaufort Gyre, the prevailing anticyclonic sea ice drift pattern that transports thick multiyear ice (MYI) from the central Arctic southward into the Beaufort Sea and beyond. Between December 2020 and February 2021, the enhanced Gyre circulation had flushed much of the Beaufort Sea's first-year ice (FYI) westward into the Chukchi Sea, simultaneously replenishing the Beaufort sea ice pack with an anomalously large ice volume import from the central Arctic (Moore et al., 2022).

By early March of most years, a wide swath of FYI extends along the Alaskan coast, providing a buffer between the Beaufort Sea's MYI pack and the southernmost reaches of the Beaufort Sea (Figure 1a). In 2021, the MYI pack had drifted further south than usual into the coastal southern Beaufort Sea (SBS). Due in part to an overall shrinking Arctic MYI area, this anomalous southward MYI flux caused a record fraction of the Arctic's March MYI pack to be sequestered in the SBS region by 1 March, 2021 (Figure 1c), based on the 1984-2022 age record (Tschudi et al., 2019b, 2020). Comprising roughly a third of 1 March sea ice in the SBS on average between 1984-2022, MYI comprised more than half of the SBS ice cover in 2021.

The presence of MYI in the SBS in March of 2021 placed a large swath of the Arctic's perennial ice pack at risk of melt. The SBS region typically sees the earliest sea ice melt onset in the Beaufort Sea, occurring in April or May (Bliss, 2023). Melt begins even earlier in the warmer Pacific-influenced waters of the Chukchi Sea, towards which ice in the SBS region usually drifts (Serreze et al., 2016; Bliss, 2023). Over the last several decades the Beaufort Sea has seen increasing melt of MYI in summer, with annual losses of MYI in the region quadrupling between 1997 and 2021 (Kwok and Cunningham, 2010; Babb et al., 2022). Since 2007, accelerated melt has become particularly intense in the SBS, which now sees substantially greater melt rates than areas north of 75 degrees (Mahoney et al., 2019). Due to this spatial variability in rates and timing of ice melt, the fate of MYI residing in the SBS at the end of winter depends on its subsequent drift through the beginning of the melt season.

Skillful prediction of sea ice transport over several-month timescales remains difficult (Reifenberg and Goessling, 2022). Winds, ocean currents, and internal ice stresses are the key governing terms in the sea ice momentum balance at time scales of days to months (Leppäranta, 2011). Winds are the primary driver of sea ice drift on these time scales (Thorndike and Colony, 1982), with ocean water typically acting as a drag force opposing the wind-driven ice motion. In a fragmented summer ice cover, internal stresses are negligible and ice motion is well predicted by winds. This zero-stress ("free drift") ice motion regime is responsible for the rule of thumb described by Nansen (1902), sometimes called the $2\%$ rule, which states that on average Arctic sea ice drifts at $1.8\%$ the wind speed and 28 degrees to the right of the surface wind direction. In the compact sea ice cover of winter and spring, however, internal stresses can be of equivalent magnitude to forcing from winds (Hunkins, 1975; Steele et al., 1997). Internal stresses are often transmitted from far-field sources, complicating predictions of ice drift from local wind forcing. Internal stresses are especially large during the consolidated season in the Beaufort Sea, the period from midwinter to early spring when the Beaufort ice cover consolidates (reaches SICs near $100\%$) and remains mostly contiguous with the Alaskan and Canadian coasts. As the as the ice pack interacts with these coastal boundaries, large internal stresses develop. These internal stresses can be of sufficient magnitude to cause the ice pack to fracture and form wide openings called leads over timescales of hours to days. These deformation events show strong correspondence to wind field divergence (Willmes et al., 2023) and are associated with rapid changes in internal ice stresses and ice motion (Richter-Menge et al., 2002). They also contribute to the episodic nature of winter ice motion in the Beaufort Sea.

During the consolidated season, flaw and coast-originating leads recurrently open along the Alaskan and Canadian coasts of the Beaufort Sea (Eicken et al., 2006; Lewis and Hutchings, 2019; Jewell and Hutchings, 2023). Coastal lead development is constrained by the position and persistence of atmospheric systems relative to the sea's coastal boundaries, with most leads forming under easterly winds which move ice away from or along the coast (Eicken et al., 2006; Lewis and Hutchings, 2019;

Jewell and Hutchings, 2023). Spatial discontinuities develop in the sea ice velocity field as coastal leads open, with greater rates of wind-driven ice drift in the lee of the lead patterns. This produces a temporary asymmetry in the rate of ice transport across the Beaufort Sea dependent upon where and when leads form (Lewis and Hutchings, 2019; Jewell et al., 2023). Given the recurrence of these events, the patterns of ice motion associated with coastal lead opening events are thought to be a primary contributor to net winter and spring ice transport. However, confirmation of this point remains challenging. In situ

measurements of sea ice drift are sparse in space and time, and satellite-derived sea ice drift products, though consistently available, typically offer limited resolution for characterizing these short-lived, localized events. Thus, precisely how the patterns of sea ice drift that occur during lead opening events contribute to seasonal ice transport has yet to be directly quantified from observations.

In early March 2021, a network of GPS ice tracker buoys was deployed as part of the Sea Ice Dynamic Experiment (SIDEx)

(Hutchings et al., 2023) in the predominantly MYI-covered SBS region. The SIDEx buoys tracked the advection of the MYI through the end of the consolidated season in late April 2021 (Fig. 2), when the Beaufort ice pack began melting and retreating. MYI transport across the SBS was episodic throughout this period, partitioned into rapid drift events punctuated by near-stationary conditions (Fig. 3b). In this analysis, we evaluated the relationship between regional atmospheric forcing, opening of wide leads detectable from remote sensing data, and the short (hourly) and long term (seasonal) patterns of ice motion in

the SBS. We analyzed ice motion between 1 March and 30 April, 2021, tracking the transport of MYI over these two months and assessing how coastal lead opening events contributed to ice transport over this period.

The layout of the paper is as follows. In Section 2, we present the data and methods used to identify coastal lead opening events, and evaluate atmospheric forcing and ice motion in the SBS region. In Section 3, we present and discuss our results. We describe the atmospheric forcing that drives coastal lead opening events, evaluate coincident changes in efficiency of wind-

driven ice drift, and quantify their role in ice transport at seasonal timescales in 2021. We also present a detailed analysis of an opening event during which coastal leads transected the SIDEx buoy network and provided direct observations of changes in dynamic ice regimes during coastal lead opening. In Section 4, we summarize and offer concluding remarks.

## 2  Data and Methods

### 2.1  Identifying lead openings from sea ice concentration data

The most frequent lead opening during the consolidated season in the Beaufort Sea is concentrated along the sea's southern coastal boundaries (Willmes and Heinemann, 2016). Leads are repeatedly opened along the border between the mobile pack ice and the semi-stable landfast ice that parallels the Alaskan and Canadian coastlines (Eicken et al., 2006; Lewis and Hutchings, 2019). Flaw leads open along the landfast ice edge, approximately parallel to the coastline. A particularly frequent flaw lead opens between Herschel Island and the Baillie Islands as part of the Cape Bathurst Polynya complex (Barber and

Hanesiak, 2004), producing a notable reduction in average winter and spring sea ice concentration (SIC) values in this area (Fig. 2). Coastal-originating leads extend offshore into the interior ice pack from various coastal promontories, approximately

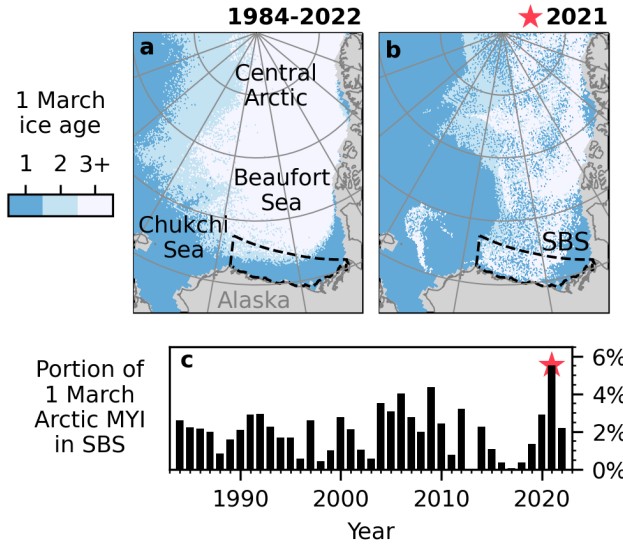

**Figure 1.** Comparison of MYI coverage in the Beaufort Sea on 1 March 2021 to the 1984-2022 record. (a) Map of median 1 March ice age (in years) between 1984-2022. (b) Map of 1 March 2021 ice age. Black dashed border outlines coastal southern Beaufort Sea (SBS) region. (c) Percentage of Arctic MYI area north of $60°$N residing in the SBS on 1 March from 1984 to 2022. Record high of $5.5\,\%$ in 2021 is nearly three standard deviations about the mean value of $2\,\%$.

perpendicular to the local coastline (Lewis and Hutchings, 2019). Point Barrow (Fig. 2) is a common activation point for these patterns.

We identified opening of coastal and flaw leads in the southern Beaufort Sea using a high-resolution daily SIC product by
Ludwig et al. (2020). The product merges 6.25 km spatial resolution passive microwave measurements from the Advanced Microwave Scanning Radiometer 2 (AMSR2) instrument aboard JAXA's GCOM-W1 satellite with 1 km cloud-free thermal infrared measurements from the Moderate Resolution Imaging Spectroradiometer (MODIS) instrument aboard NASA's Aqua satellite. The final merged product, which we will refer to as MODIS-AMSR2 SIC, has 1 km spatial resolution. It therefore can be used to detect opening of wide leads that are missed in products with coarser spatial resolution.

We focused on identifying lead openings within the coastal southern Beaufort Sea, henceforth referred to as the Southern Beaufort Sea (SBS) region. The SBS region covers an area of $2.3 \times 10^5\ \mathrm{km}^2$ extending along the Alaskan coast, bounded in the west by the Beaufort-Chukchi Sea border and in the east by the Canadian coast (Fig. 1). We used MODIS-AMSR2 SIC data to estimate the area containing open leads within the SBS region. We define the open extent within the SBS region as the area sum of $1\ \mathrm{km}^2$ grid cells that contain open water concentrations of more than $20\,\%$. Ludwig et al. (2020) find uncertainties
of $5 - 10\,\%$ for their SIC estimates between the months of February and April. This could lead to erroneous detection of open water concentrations of up to $10\,\%$ for sea ice that is actually completely compact (SIC $= 100\,\%$). Thus, we set a minimum $20\,\%$ open water concentration threshold to only include openings detected with a high degree of confidence, i.e. those with SIC values well outside the range of potential SIC estimates for compact sea ice. Thin ice in frozen-over leads may reduce

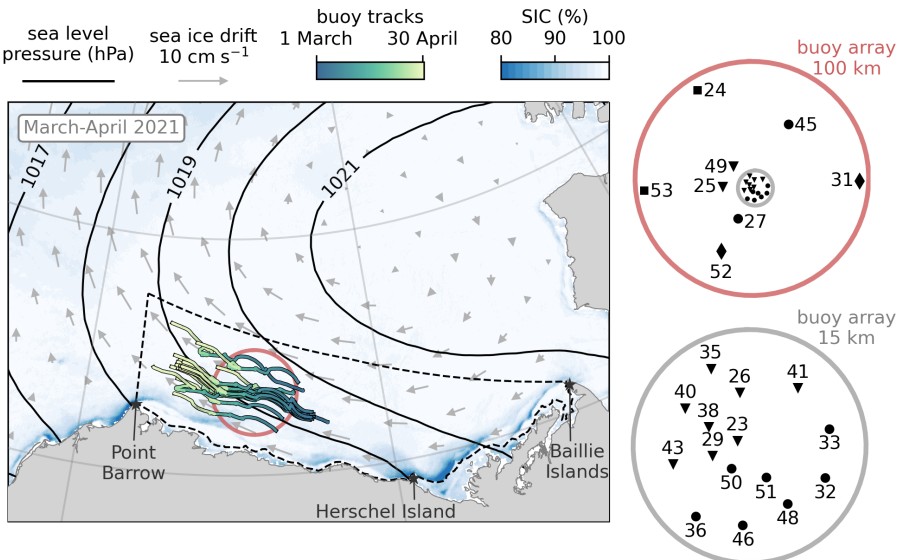

**Figure 2.** Regional conditions and SIDEx buoy drift during March-April 2021. Mean sea level pressure (black lines) and Polar Pathfinder sea ice drift (grey vectors) overlain above the mean SIC field. SIDEx buoy tracks from 1 March - 30 April overlain. Dashed line shows SBS region border. Red circle shows position of 100 km buoy array on March 20, when all buoys were deployed. Red and grey circles to the right of the map show buoy positions within the nested 100 km and 15 km buoy arrays. Buoy symbols correspond to dynamic aggregates observed in case study opening event.

estimated SIC values despite being ice-covered. However, this was acceptable for this analysis as we aimed to detect recently
opened leads, which quickly form a thin ice cover in freezing conditions.

In many SIC products, including the MODIS-AMSR2 product used here, erroneous SIC reductions often occur over areas of contiguous landfast ice throughout winter (see, for example, the reduced SIC southeast of Herschel Island in Fig. 2). Landfast ice typically extends offshore to $20-30\,\mathrm{m}$ depth throughout the SBS region in winter (Mahoney et al., 2014; Nghiem et al., 2014; Trishchenko et al., 2022). We used bathymetry data from GEBCO Compilation Group (2022) to bound the SBS region
where it meets the coast by the $10\,\mathrm{m}$ isobath, slightly inshore from the landfast ice edge. This boundary omitted most erroneous SIC estimates within the landfast ice area but included openings along the landfast ice edge. Directly east of Herschel Island, we bounded the SBS region directly along the March-April 2021 stable landfast ice edge since it extended far past the $10\,\mathrm{m}$ isobath and frequently showed erroneous SIC values in 2021.

For a case study of one opening event, we acquired Level-1B thermal infrared MODIS imagery (MODIS Characterization
Support Team (MCST), 2017a, b) and Visible Infrared Imaging Radiometer Suite (VIIRS) imagery (VCST Team, 2021a, b) to more precisely identify the timing and location of lead openings. We used MODIS Band 31 ($10.78-11.28\,\mu m$) which has 1 km resolution and VIIRS Band M15 ($10.26-11.26\,\mu m$) which has 750 m resolution.

For a longer-term analysis of lead opening in the SBS, we used the 6.25 km SIC product by Spreen et al. (2008), which is based primarily on the 89 GHz channel from AMSR-E and AMSR-2 and is available from 2002 - present. This product is one of the two used in the merged MODIS-AMSR2 product (available 2017-present), and therefore yields very similar estimates of SIC. We compared estimate of SBS open water extents derived by the two products over March - April 2021, and found a daily mean absolute difference of just $3\,\mathrm{km}^2$.

## 2.2 Defining consolidated sea ice season duration and opening events

We first used the calculated open water extent in the SBS region to determine what portion of winter and spring 2020-2021 fell within the consolidated sea ice season in the Beaufort Sea. We define the Beaufort Sea consolidated season as the period when the Beaufort ice pack is compact (reaches SIC near $100\,\%$) and remains mostly contiguous with the the landfast ice and coastlines bounding it. This period begins in early winter when dynamic and thermodynamic consolidation causes the ice cover to expand southward, meeting the Alaskan and Canadian coastlines. Throughout the consolidated season, the ice pack episodically pulls away from the coast along flaw leads and coast-originating leads, which dynamically close or freeze over within days. The end of the consolidated season usually begins in early May when southeasterly winds open the recurrent winter flaw lead along the Cape Bathurst Polynya complex, but elevated temperatures prevent the flaw lead from freezing over (Galley et al., 2016; Huang et al., 2022). After this dynamic spring breakup begins, dynamic and thermodynamic processes cause the ice to continue retreating northward for the remainder of the season.

We defined the consolidated season to extend from the first to the last day of the 2020-2021 growth season when the SBS open extent (total area where SIC $< 80\,\%$) dropped below $1000\,\mathrm{km}^2$. With this definition, the consolidated season extended from 20 December, 2020 until 25 April, 2021. Within this four month period, the sea ice cover episodically re-opened along leads in areas exceeding $1000\,\mathrm{km}^2$ for 1-5 days at a time (Fig. 3a). We refer to these days as "open days," and to periods of one or several consecutive open days as consolidated season lead opening events. We refer to days when the area with lead openings remained below $1000\,\mathrm{km}^2$ as "compact days."

## 2.3 Analyzing winds and sea ice motion in the SBS region in 2020-2021

We used the $25\,\mathrm{km}$ resolution satellite-derived Polar Pathfinder sea ice drift product (Tschudi et al., 2019a) to provide a general description of ice drift throughout the SBS region over the entire 2020-2021 sea ice growth season. For each day, the average sea ice velocity vector where ice was present in the SBS region was calculated. The speed of this mean daily drift vector is shown in Fig. 3b. To analyze hourly 10 meter winds and mean sea level pressure, we used ERA5 (Hersbach et al., 2023), the fifth generation climate reanalysis from the European Centre for Medium-Range Weather Forecasts (ECWMF), which has $31\,\mathrm{km}$ ($0.25°$) spatial resolution. ERA5 winds were spatially-averaged in the SBS region (Fig. 3c) to describe general patterns of atmospheric forcing. Both ERA5 atmospheric conditions and Polar Pathfinder drift fields were used to create maps of synoptic-scale conditions across events.

Beginning in late February 2021, SIDEx buoys were placed throughout the area of MYI residing in the SBS region (Hutchings et al., 2023). Throughout March, additional buoys were deployed to form a network of buoys arranged in arrays with radii

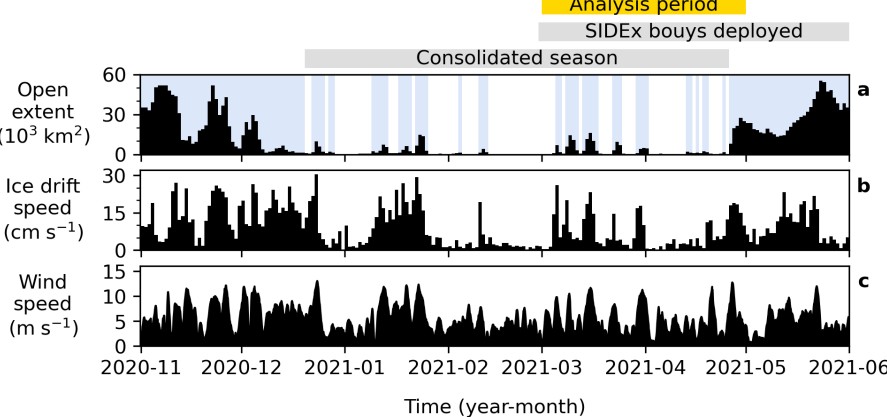

**Figure 3.** Overview of SBS sea ice and atmospheric conditions during winter and spring 2020-2021. Relevant time periods shown above panels. (a) Daily open water extent in the SBS region, with periods $> 1000 \, \text{km}^2$ shaded blue. (b) Daily sea ice drift speed calculated from Polar Pathfinder drift averaged across ice area in the SBS region. (c) Speed of hourly winds averaged across the SBS region.

ranging from $15 - 100 \, \text{km}$ (Fig. 2). Over the following months until the melt season began at the end of April, they tracked the transport of ice from the SBS toward the boundary of the Beaufort and Chukchi Seas. We utilized these lagrangian measurements of sea ice drift to directly compare sea ice motion throughout the SIDEx network to local wind forcing at hourly timescales. The buoys provided direct measurements of temporal and spatial patterns of sea ice drift during consolidated season lead opening events.

The SIDEx buoys reported latitude and longitude coordinates at 10 minute intervals (Hutchings et al., 2023). Buoy drift tracks were de-spiked then interpolated to 30 minute time steps using cubic splines prior to computing hourly velocities (Fig. 3c). We analyzed 23 of the buoys in the network, omitting nine buoys that were either deployed later than 20 March, removed before April 30, or were within $2 \, \text{km}$ of another buoy throughout the analysis period. When buoy motions were directly quantitatively compared to winds, we used linear interpolations of the gridded ERA5 data to estimate the wind vectors directly above each buoy.

## 3    Results

### 3.1    Overview of atmospheric and ice conditions in March-April 2021

Figure 4 shows a time series of sea ice drift speeds measured by SIDEx buoys throughout March and April 2021 in relation to SBS atmospheric forcing and the open extent in the SBS region. Figure 4a shows the timing of consolidated season opening events, which spanned 29 of the 61 days between 1 March and 30 April ($48 \, \%$ of days). These events were partitioned into ten events lasting 1-5 consecutive days labeled sequentially as events E1 - E10 in Fig. 4. During the other 32 days between 1 March and 30 April ($52 \, \%$ of the time), the ice remained compact. Compact periods lasted 1-11 consecutive days.

Note that just two SIDEx buoys were present when event E1 began, and the remaining buoys were deployed between events E1 and E3 (Fig. 4d). Thus, more comprehensive measurements of the spatial patterns of sea ice drift throughout the region covered by the $100\,\mathrm{km}$ buoy array were possible for events E4 and on. The final event (E10) beginning 26 April marked the start of spring breakup and therefore the end of the consolidated season in the Beaufort Sea. After event E10, the SBS open extent did not fall below $1000\,\mathrm{km}^2$ until the following winter.

Panels (b-e) in Fig. 4 reveal some general contrasting features in the atmospheric conditions and ice drift between open and compact days throughout March and April 2021. Opening events tended to coincide with periods of sustained easterly or southerly wind forcing. These wind orientations blow ice away from the Beaufort Sea's south and east coastal boundaries, and toward westward seas with more distant coastlines. Wind speeds and ice drift speeds (as measured by the SIDEx buoy network) tended to increase as the ice opened, then decrease as it closed. Most local maxima in SBS open extent roughly matched the timing of local maxima in wind speeds (mostly ranging $10-15\,\mathrm{m\,s^{-1}}$) and ice drift speeds (mostly ranging $30-40\,\mathrm{cm\,s^{-1}}$). As a result, opening events were usually associated with elevated wind speeds and sea ice drift speeds. Bands of especially rapid ice drift occur during during some opening events, usually corresponding to progressive fracturing of the ice pack. For example, event E5 saw multiple periods of especially rapid drift lasting several hours each. Satellite imagery in Movie S1 and Movie S2 (supplemental material) shows these correspond to opening of coastal leads upwind of the SIDEx buoy network. In contrast, wind speeds and ice drift speeds were generally lower during the compact days between opening events. These periods predominantly featured westerly wind forcing, blowing ice toward the Beaufort Sea's eastern boundary. There were several sustained compact periods (e.g. 27-28 March, 1-3 April) during which the ice around the SIDEx buoys remained completely stationary.

The close correspondence between changes in atmospheric conditions and transitions between open and compact days indicates that regional atmospheric weather conditions were a dominant control on the timing and progression of opening and closing events. The beginning, peak, and end of many opening events coincided with changes in atmospheric forcing shown in Fig. 4(b-d). Specifically, transitions between open and compact periods often coincided with abrupt changes in the orientation of wind forcing in the SBS region. These changes in wind orientation typically corresponded to local extrema in sea level pressure, which signal changing weather conditions. For example, the especially long and large opening events (E1-E5) began shortly after peaks in sea level pressure indicating the passage of high pressure systems over the SBS region, causing a shift from northerly or westerly winds to easterly winds. The peaks and ends of these opening events also roughly coincided with local atmospheric pressure extrema.

The period in mid-to-late April between events E6 and E9 presents some exceptions to the above summarized relations between wind orientation, opening, and resultant ice drift speed. For example, ice drift and wind speeds are relatively low during events E6 - E9. These opening events were shorter and more localized than events occurring in March (see Fig. 5). Another interesting exception is when the ice drifted rapidly under westerly winds during a compact period on April 20. Earlier in the season (March and early April), westerly wind events did not cause rapid ice drift. However, by this time, the ice pack had begun to become less compact as the end of the consolidated season neared. Furthermore, during event E8, occurring just the day before, the ice had pulled away from the eastern boundary of the SBS. Thus, for one day, strong westerly winds

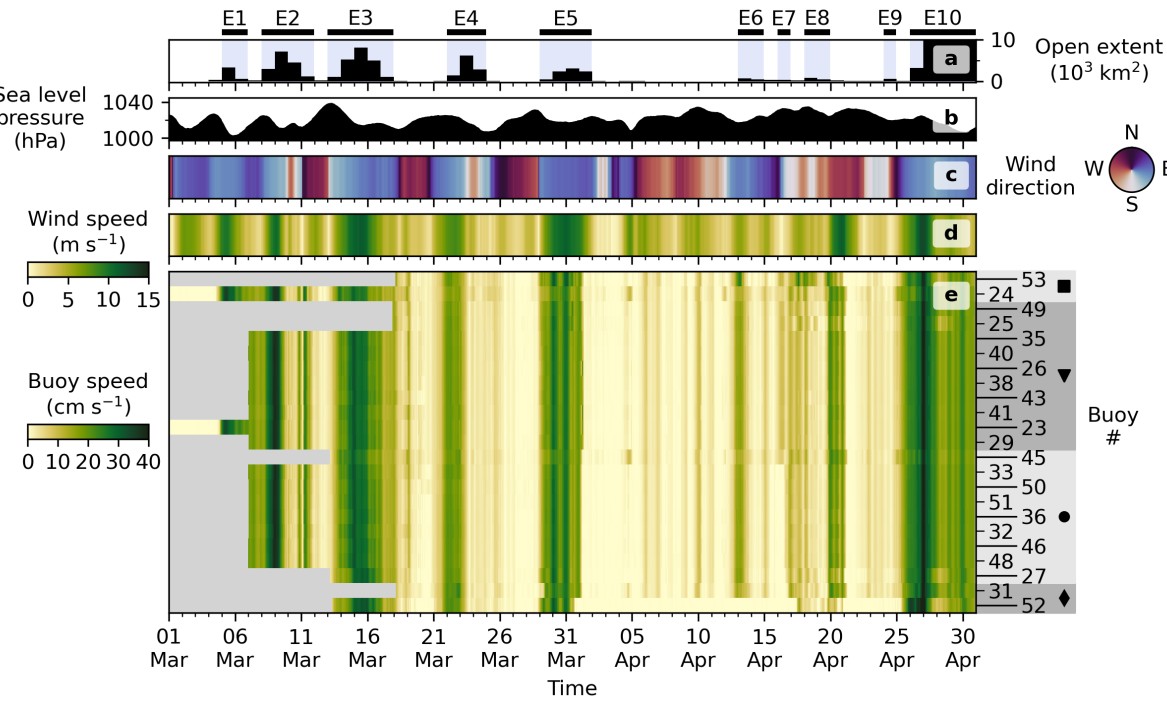

**Figure 4.** SBS conditions during March-April 2021. SBS open extent (a), SBS sea level pressure (b), SBS wind direction (c), SBS wind speed (d), and SIDEx buoy drift speeds (e). Opening events labeled E1 - E10 above figure and highlighted in blue in panel (a). Grey area in panel (e) indicates no data. Buoys grouped by positions using symbols from Fig. 2.

were able to push the pack ice pack eastward toward the Beaufort Sea's eastern boundary before it reconsolidated. Further
insight into the different conditions across opening events is provided in the following section where we present spatial maps
of each opening event.

## 3.2   Synoptic atmospheric forcing of coastal lead opening events

We next generated spatial maps of the synoptic atmospheric forcing conditions and patterns of ice drift during consolidated
season lead opening events, as well as the periods when the ice remained compact. All identified opening events were associated
with high pressure centers or ridges located north or east of the SBS region, producing pressure gradients that drove southerly or
easterly winds over some or all of the SBS region. The positions and shapes of the pressure systems varied, as did the resultant
patterns of lead opening in the SBS region. Generally, opening occurred in the SBS region wherever pressure gradients were
oriented parallel to the adjacent coastline (on the maps, where pressure contours were perpendicular to the coast), producing
winds that drove the ice away from local coastal boundaries. For additional insight into connections between opening patterns
and forcing, supplemental Movies S1 and S2 show atmospheric forcing, SIDEx buoy drift, and thermal infrared satellite
imagery of the ice cover at 4-hourly time steps between 1 March and 30 April.

Of the ten opening events, eight (E1-E6, E8, E10) were associated with opening of flaw leads along the eastern boundary of the SBS region. These were the most pronounced events, all lasting multiple days and contributing some of the most rapid drift over the analysis period. Each of these events except E8 featured a strong pressure gradient driving offshore winds between Herschel and Baillie Islands, between which the flaw leads extend. During most events the SIDEx buoys measured rapid ice acceleration as the flaw leads opened 300-800 km southeast of their positions, indicating coherent ice motion throughout the SBS region. During events with the greatest areas of opening (E1-E5, E10), the ice around the SIDEx buoys drifted especially rapidly northwestward as the entire SBS ice pack moved along the Alaskan coast.

Most events also featured opening near the western portion of Alaska's northern coast. These either took the form of flaw leads (E1-E3, E10) or coast-originating leads that extended offshore into the central ice pack (E6, E7). Event E9 also featured a coast-originating lead in the eastern SBS extending offshore from the typical flaw lead location. Events E7 and E9 were the only events in which flaw leads were not detected. They were also the only single-day events. This is likely a result of the overall weaker and less spatially coherent pressure gradients throughout the SBS region during these periods. Strong alongshore pressure gradients (offshore winds) only occurred local to the regions where the coastal-originating leads opened, in the westernmost (E7) and easternmost (E9) portion of the SBS.

The average conditions across all compact days, comprised of ten distinct sequences, is shown in Fig. 5k. Compact days were on average associated with a weak high pressure system positioned over the western SBS region. Resultant weak winds drove ice toward the Beaufort Sea's eastern and southern coastlines, and the ice pack remained mostly stationary.

### 3.3 Contribution of ice drift during opening events to seasonal transport

Together, Fig. 4 and Fig. 5 show that opening events were associated with rapid rates of sea ice transport, moving MYI located in the central SBS region along the Alaskan coast toward the west-northwest. This direction of transport is typical for sea ice in the SBS region during the consolidated season. To understand how ice drift patterns during lead opening events contribute to ice transport at seasonal timescales, we next quantified how much of the cumulative displacement of the SIDEx buoys in March and April 2021 occurred during detected lead opening events.

We focused on Buoy 23, one of the first buoys deployed and one of two buoys that were present for the entire analysis period. Buoy 23 was located near the center of the SIDEx buoy network within the 15 km buoy array (Fig. 2). We calculated the cumulative displacement of Buoy 23 to estimate the overall displacement of the ice surrounding the SIDEx buoy network between 1 March and 30 April. The average hourly drift of Buoy 23 over this period was oriented toward 67.2° west of north, almost exactly along the west-northwest (WNW) direction (67.5° west of north). We therefore took the WNW direction as the mean direction of travel, and for each hourly displacement we calculated the component of the hourly drift along the WNW direction.

In total, Buoy 23 traveled a net $365\,\mathrm{km}$ along the WNW direction between 1 March and April 30 (Fig. 6c). Ice motion was strongly constrained to this direction, with only $2\,\mathrm{km}$ net displacement occurring perpendicular to the WNW direction over the same period. The rapid WNW ice transport was episodic, partitioned mostly into several-day bursts of rapid transport

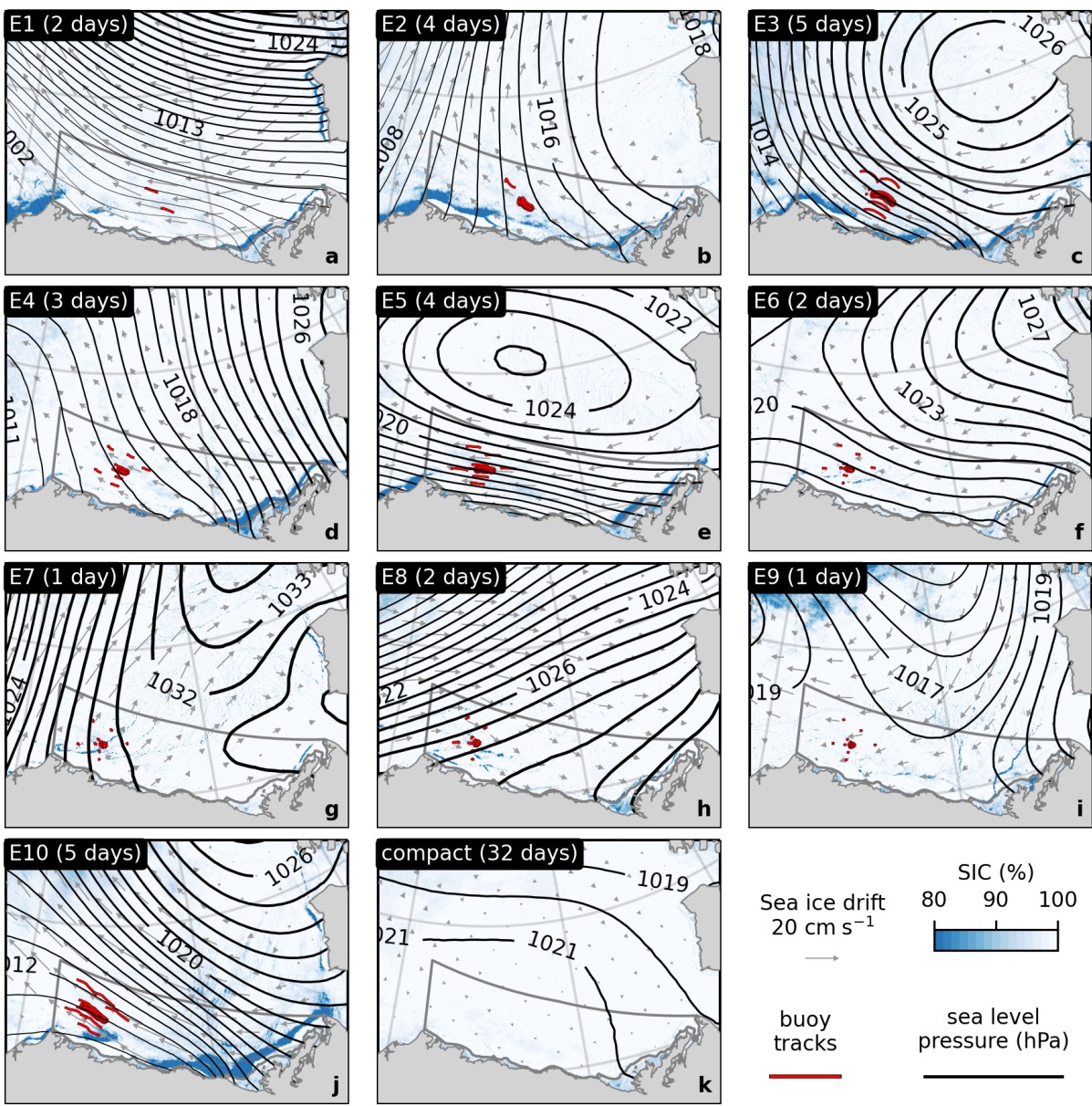

**Figure 5.** Comparison of March and April coastal lead opening events with compact periods. Mean SIC fields are overlain with Polar Pathfinder sea ice drift (grey vectors) and sea level pressure contours (line width increases with pressure). For each opening event E1 - E10 (a-j), conditions are averaged across all days in the event period. Buoy drift tracks from event start to end are shown in red. Average conditions for all 32 compact days shown in panel (j).

corresponding to detected opening events. Periods of rapid transport were punctuated by several-day compact periods with near-stationary conditions.

Figure 6c also compares the net seasonal WNW transport to that which occurred just during the detected lead events. The net WNW displacement during lead opening events was $343\,\mathrm{km}$, accounting for $94\,\%$ of the total. Thus, nearly all of the seasonal transport occurred during periods when leads were detected in the SBS ice pack, which spanned less than half of the total days. Compact days contributed just $22\,\mathrm{km}$ along the WNW direction, accounting for $6\,\%$ of the total. With some drift reversals and other off-WNW displacements, the network traveled a total distance of $498\,\mathrm{km}$. Open days accounted for $79\,\%$ of this distance.

Figure 6(a,b) shows the relations between buoy-measured ice drift and local winds, quantified by the ratio of ice drift to wind speeds and the turning angle between wind and ice drift directions. These data demonstrate that open days typically featured enhanced ratios of ice drift speeds to wind speeds compared to compact days. Changes in speed ratio between these conditions was often abrupt, occurring in just a matter of hours. This effect was most clear for the most pronounced opening events, which occurred in March. Event E5, for example, was found to begin shortly after the ice-to-wind speed ratio increased abruptly from near zero to around $2\,\%$ on 28 March. The speed ratio hovered steadily around $2\,\%$ for several days, then abruptly returned to near-zero conditions on the last day with detected openings. Such transitions were less clear for the shorter-lived and more localized opening events occurring in mid April (e.g. event E9). Another key distinction between open and compact days was the difference in wind-ice turning angles. On compact days, as speed ratios typically fell to near zero, wind-ice turning angles were highly variable. During pronounced opening events, increases in speed ratio coincided with stabilization of wind-ice turning angles between $0° - 30°$. Though internal ice stresses were not directly measured, the changes in wind-ice turning angles and speed ratios between open and compact periods are indicative of the ice transitioning back and forth between free drift and high ice-interaction conditions.

## 3.4 Difference in dynamic regimes between open and compact periods

Though qualitative, the described differences in wind-ice relations suggest differing dynamic ice regimes are responsible for the rapid (weak) wind-driven ice transport on open (compact) days. To investigate these differences further and understand why nearly all seasonal transport occurs during opening events, we next quantified the differences in wind and ice conditions between open and compact days. We also expanded this portion of the analysis to include all buoys for all hours they were present during the analysis period. Figure 7 shows distributions of buoy drift speeds and directions, as well as wind speeds and directions local to the buoy positions. These distributions are provided for all hourly observations on open days (N=13834), and separately for all observations on compact days (N=15296).

These data quantitatively confirm the large difference in rates of ice drift between open and compact days. The median ice drift speed across open days ($15.8\,\mathrm{cm\,s^{-1}}$) was nearly an order of magnitude greater than the median speed when the ice was compact ($1.9\,\mathrm{cm\,s^{-1}}$). Considering that winds are the primary drivers of ice motion, it is notable that the median wind speed only increased by $60\,\%$ from compact days ($3.9\,\mathrm{m\,s^{-1}}$) to open days ($6.1\,\mathrm{m\,s^{-1}}$). For ice moving in free drift, not under the action of internal ice stresses, ice drift speed should increase linearly with wind speed (Park and Stewart, 2016). Thus,

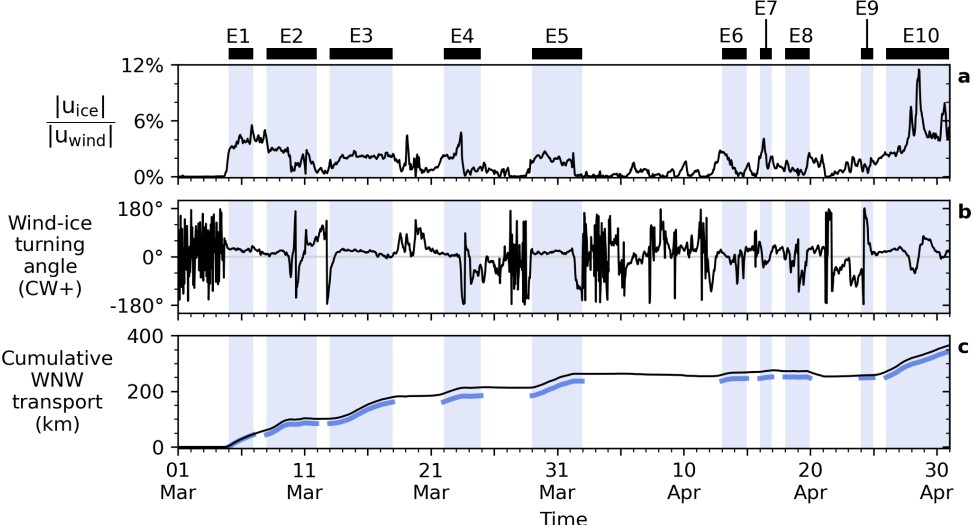

**Figure 6.** Drift characteristics of buoy 23 throughout March and April 2021. (a) Hourly ratio of buoy speed ($|U_{ice}|$) to local wind speed ($|U_{wind}|$). (b) Hourly wind-to-ice turning angle (clockwise turning is positive). (c) Cumulative displacement from 1 March, 2021 (00:00 UTC) along the WNW direction (azimuth $-67.5°$, approximately the average direction of travel). Black line shows total displacement, while blue curve shows displacement occurring during opening events only. In all panels, blue shading indicates days with satellite-detected lead opening events.

the superlinear increase in the rate of ice drift relative to winds suggests a change in the sea ice momentum balance, likely a reduction in internal ice stress as the ice opens.

We then directly compared local winds and ice drift in panels (e) and (f) of Fig. 7. These data make a particularly strong case for the varying role of internal ice stresses between open and compact periods, and their subsequent impacts on rates of ice transport throughout March and April 2021. During open days, most data were tightly clustered around a narrow range of values in the two-dimensional distribution of ice-to-wind speed ratios and wind-ice turning angles. More than $40\%$ of all measurements fell with the range $1.5\% - 3\%$ for the speed ratio, and $0° - 30°$ for the wind-ice turning angle. Most frequently, the ice drifted at $2.2\%$ the wind speed and $20°$ to the right of the wind direction. These are within typical ranges for ice in free drift (Leppäranta, 2011; Brunette et al., 2022). In contrast, compact days were characterized by a diffuse distribution of wind-ice relations. The median turning angle, $15°$, was similar to that of open days, though overall turning angles were much more broadly distributed. Ice-wind speed ratios were primarily concentrated around the lowest values, with a median of $0.6\%$. Fewer than $10\%$ of measurements fell within the range of free drift values previously described.

Finally, we summarize these results in Fig. 8. There we show the frequency of ice drift rates measured by all SIDEx buoys throughout March-April 2021 as a function of wind direction over the SBS region. We also highlight the SBS wind directions that were particularly frequent during open and compact ice conditions in the SBS region. Altogether these observations strongly suggest that changing internal ice stresses are responsible for the differing rates of ice transport between open and

compact portions of the consolidated season. Furthermore, they highlight how the orientation of wind forcing relative to coastal
boundaries controls ice stress states and subsequent rates of ice transport.

In the SBS region, winds most frequently blew from the east-southeast direction, or from the west-northwest directions. These two primary wind orientations blow roughly parallel to the Alaskan coast at the southern boundary of the SBS region. Of these two primary wind orientations, west-northwesterly winds push ice toward the Canadian coastline that forms the eastern boundary of the SBS region. This wind direction was especially likely to produce compact periods. Even under moderate wind 310 forcing, the ice remained mostly stationary as it was pushed toward coastal boundaries, usually drifting well below $1\,\%$ of the wind speed. Internal stresses likely built within the compact ice cover, balancing wind forcing and reducing the efficiency of wind-to-ice momentum transfer.

The other most frequent wind orientation, east-southeasterly winds, blows ice away from local SBS coastal boundaries, toward regions with more distant coasts. This wind direction became especially common during opening events. As it drove 315 ice away from coastal boundaries, the efficiency of wind-driven ice transport tended to be high, with ice drifting at $2\,\%-3\,\%$ of the wind speed. This likely resulted from a reduction in internal stresses as the ice cover diverged. Ice stresses may even have fallen to zero, as has been observed during similar consolidated season opening events in the Beaufort Sea (Richter-Menge et al., 2002). As shown in Fig. 7, wind speeds also tended to be higher as the ice opens. Thus, heightened wind speeds coupled with substantially heightened rate of wind-to-ice momentum transfer, likely produced the exceptional rates of ice transport 320 during easterly-driven opening events. It is interesting to note that during many opening events the ice around the SIDEx buoys remained mostly compact as it separated from the coast. Thus, conditions characteristic of free drift appear to have been reached in areas where overall SIC remained above $80\,\%$, a threshold often suggested as the minimum ice compactness for which internal stresses remain negligible (Leppäranta, 2011).

### 3.5 Spatiotemporal patterns of sea ice motion during Event E5

Several observational studies have identified changes in the rate and efficiency of winter sea ice transport in association with opening of leads in the Beaufort Sea (Lewis and Hutchings, 2019; Jewell et al., 2023). To evaluate changes in ice motion, these studies often rely on satellite-derived sea ice drift products, which are typically available at daily temporal resolution. Given that leads can open and close in a matter of hours, it can be challenging to directly attribute detected changes in ice transport to the ice drift patterns associated with lead opening. In situ measurements of ice motion offer finer temporal resolution, but 330 are sparse. The network of SIDEx buoys analyzed here offered spatially dense measurements of ice motion at hourly intervals. Thus, they captured spatiotemporal changes in the efficiency of wind-driven ice transport as the ice transitioned between open and compact conditions.

We investigated one opening event as a case study which shows that abrupt changes in wind-to-ice momentum transfer in March-April 2021 were associated with opening and closing of coastal leads in the SBS region. We focus on Event E5, 335 during which the dense network of SIDEx buoys measured spatial discontinuities in the sea ice velocity field. We acquired thermal infrared MODIS and VIIRS imagery to visually assess the continuity of the ice cover with greater temporal precision than offered by the daily SIC data. The satellite imagery confirmed that leads had directly transected the array of SIDEx

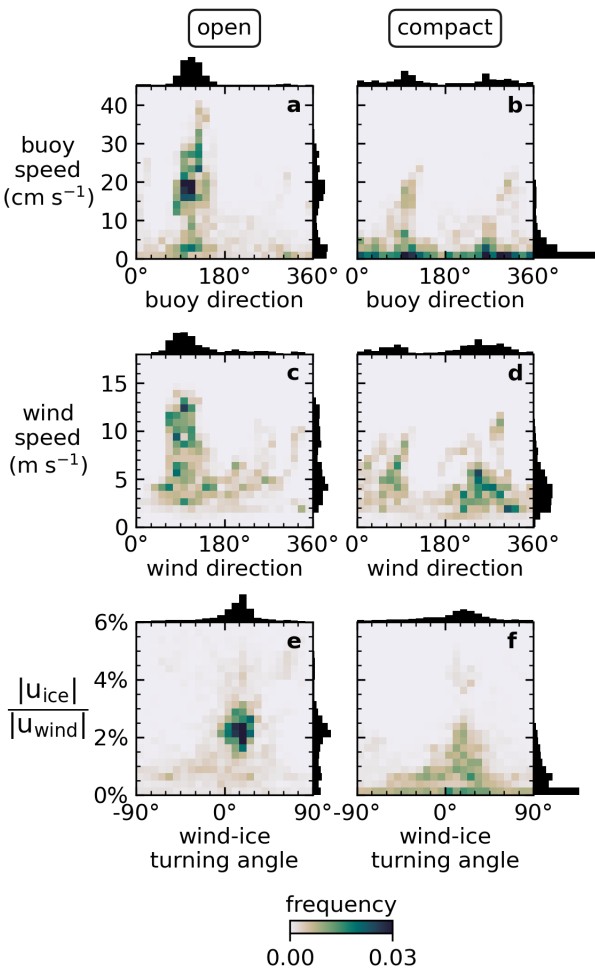

**Figure 7.** Comparison of hourly sea ice and wind conditions across open days (left column, N=15361) and compact days (right column, N=13769) of March-April 2021. Distributions of buoy drift speeds and directions (a,b), wind speeds and directions (c,d), and wind-ice speed ratios and turning angles, clockwise positive (e,f). Meteorological convention used for buoy and wind directions. Winds are interpolated to local buoy positions. Color scale shows frequency of two-dimensional distributions. Individual variable distributions shown in black, with relative frequencies scaled equivalently across panels.

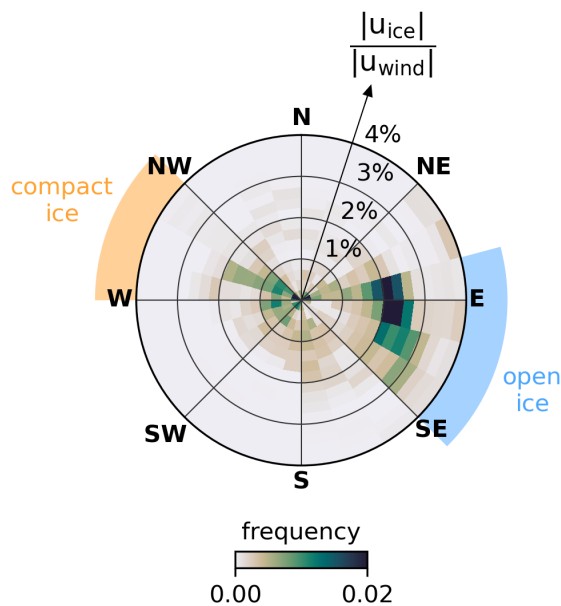

**Figure 8.** Relation between ice transport efficiency, compactness of the ice cover, and SBS wind direction in March-April 2021. Color scale shows frequencies of ice-wind speed ratios (radial axis) measured by all buoys as a function of SBS wind direction (angular axis). Wind directions that are especially frequent during compact (open) conditions highlighted in orange (blue). Wind directions highlighted for each ice condition when they occur at least $5\%$ of the time during those conditions, and are twice as likely during those conditions than the other ice conditions.

buoys where and when velocity discontinuities were measured. They also showed that rates of wind-to-ice momentum transfer abruptly increased where leads separated the once-compact ice pack from the coast. Supplemental Movies S1 and S2 include animations of thermal infrared MODIS and VIIRS imagery of ice conditions and its impacts on the relation between winds and SIDEx buoy drift during opening event E5.

Event E5 was a four-day coastal lead event detected as a reduction in SBS SIC beginning on 29 March. In situ ice velocity measurements from SIDEx buoys showed the initial opening likely began in the final hours of 28 March as a high pressure atmospheric system moved over the SBS region, causing winds to abruptly shift from westerly to easterly (Fig. 9a). Around the same time, the previously compact, motionless ice cover rapidly accelerated, causing an abrupt increase in the ratio of ice drift speeds to local wind speeds (Fig. 9b). After the initial opening, ice throughout the SIDEx-sampled region drifted coherently for several days at $2\% - 3\%$ the speed of local winds. Winds gradually developed a northerly component and steadily slowed. Between 31 March and 1 April, the ice re-consolidated with the coast and became stationary again.

SIDEx buoys measured rapid increases and decreases in buoy velocities on the days when daily SIC data showed the SBS ice pack opening and closing. During this opening and closing, the buoy network measured stepwise spatial progressions in ice velocity which produced temporary spatial discontinuities in the ice velocity field. Figure 5b shows the ratio of ice drift speeds to local wind speeds measured by the SIDEx buoy network. Wind speeds were smoothly varying across the SIDEx network (see

panels c-f in Fig. 5). Thus, the sharp spatial discontinuities in ice-wind speed ratios shown in Fig. 5b result from discontinuities in the ice velocity field. Thermal infrared satellite imagery showed the development of ice velocity discontinuities corresponded

to a series of coast-originating leads which extended across the buoy network. These leads usually became clearly visible in $1 \, \mathrm{km}$ resolution MODIS imagery or $750 \, \mathrm{m}$ resolution VIIRS imagery several hours after the velocity discontinuities were measured by buoys. This delay makes sense, given that during event E5 the average rate of WNW ice drift was $13 \, \mathrm{km \, day^{-1}}$. If a new fracture formed between the stationary ice pack and a portion of the ice pack that begin drifting at this average rate, it would take several hours for the initial fracture to widen past the minimum spatial resolution of the satellite imagery.

Beginning on 28 March, a series of coast-originating leads began extending offshore from the landfast ice around Point Barrow, Alaska. At 09:00 UTC, the two northwesternmost buoys (square symbols) began accelerating relative to local wind speeds, while the rest of the buoy network remained stationary (Fig. 9b). In a satellite image from around 12:00 UTC, a developing lead could be seen extending perpendicular to the coast, separating the two northwesternmost buoys from the rest of the network (Fig. 9c). By 13:00 UTC, all buoys but the easternmost began accelerating as another coastal lead began

developing across the network. In imagery from around 07:00 UTC on 29 March, coastal leads can be seen extending offshore across the buoy network, delineating the easternmost buoy and the rest of the buoy network (Fig. 9d).

        Before opening, the SBS ice pack was able to resist moderate ($5 \, \mathrm{m \, s^{-1}}$) westerly wind forcing, remaining stationary as the winds compressed the ice against the coastal boundaries of the SBS region. However, as winds shifted easterly over a portion of the buoy network, blowing offshore toward an already-fractured ice cover, the ice quickly opened and began drifting. This

initial progression of lead opening occurred as wind speeds were dropping to near-zero over the center of the buoy network (Fig. 9a), demonstrating the weak tensile strength of the sea ice cover. For two and a half days, the entire buoy network drifted coherently at approximately $2 \, \% - 3 \, \%$ of local wind speeds as the mobile SBS ice pack slid northwestward along the Alaskan coast (Fig. 5e). Bands of enhanced ice drift speeds (and ice-to-wind speed ratios) occurred during this time as additional fractures opened and developed upwind of the bouy network (visible in Movies S1 and S2), culminating in opening of a flaw

lead along the eastern boundary of the SBS region. Few fractures were visible directly in the vicinity of the SIDEx buoy network as the mobile ice pack sheared along the coast (Fig. 9e).

        By 17:00 UTC on 31 March, the southernmost SIDEx buoy rapidly slowed as it re-consolidated with ice that had remained fastened to the coast throughout the opening event. By 06:00 UTC on 1 April, the entire southeastern portion of the buoy network had abruptly stopped drifting. Buoys in the northwest continued drifting at $2 \, \%$ the wind speed for several hours until

they too abruptly immobilized. Cloud cover partially obscured the ice at this time. However, imagery from the following day (Fig. 9f) shows a coastal lead characteristic of a compressive shear fault (Schulson, 2004) directly transecting the central buoy array where the velocity discontinuity had persisted for several hours. This indicates that the stepwise immobilization of the ice cover occurred as the ice was progressively re-consolidating with the landfast ice along the coast. The development of a lead across the central buoy array allowed the southeast and northwest sections of the buoy network to move as distinct dynamic

aggregates for several hours, with the northwest portion continuing to drift efficiently in response to winds while the southeast portion made contact with the coast.

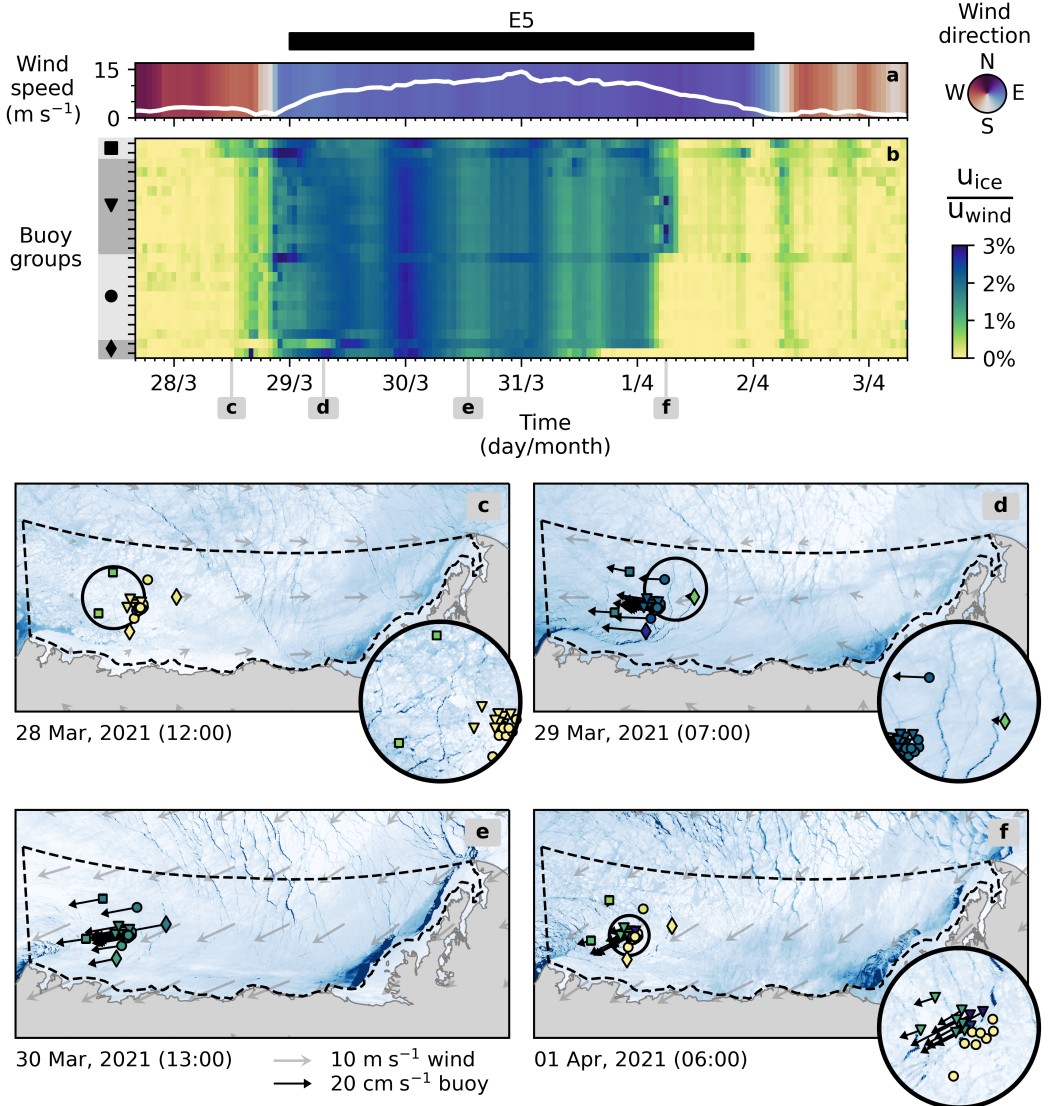

**Figure 9.** Spatiotemporal patterns of sea ice drift during opening event E5. (a) Time series of wind speed and direction around buoy 23, near the center of the buoy network. (b) Ratios of buoy speeds to local wind speeds. Buoys are listed vertically in same order as in Fig. 4, and are separated into groups marked with symbols corresponding to buoy positions in lower panels (c-f). (c-f) Snapshots of thermal infrared MODIS and VIIRS imagery, wind forcing (light arrows), and buoy drift (dark arrows) during hours highlighted in panel (b). Colors of buoy markers shows ice-wind speed ratios matching scale of panel (b). Dark (light) shades in imagery show open leads (consolidated ice). Imagery from within one hour of listed time, except for panel (f) where image is chosen for day after listed time when cloud cover reduced. Ice conditions changed little over this period.

## 3.6 Consolidated season dynamics in the SBS over the last two decades

SIDEx buoy measurements supported detailed investigations of the correspondence between wind forcing, coastal lead opening, and the rate of sea ice drift in the SBS in March - April 2021. Environmental conditions in the Beaufort Sea have changed considerably over recent decades, modulating the dynamics of the consolidated ice cover. To understand interannual and longer variability we extended our analysis across additional years (2003-2024) and all months of the SBS consolidated ice season (January-April). For this we used coarser-resolution remotely sensing data to determine how conditions in 2021 compared to other years.

First, we extended the detection of coastal lead events from January-April 2003-2024 using the $6.25\,\mathrm{km}$ SIC product from the University of Bremen (Spreen et al., 2008). We compared estimates of SBS open water extent over March-April 2021 using this product and the higher-resolution MODIS-AMSR2 product, and found a daily mean difference of less than $3\,\mathrm{km^2}$. With this small difference, we considered this product comparable to MODIS-AMSR2 and used it to identify lead opening events with the same definition as before (days with open water extents exceeding $1000\,\mathrm{km^2}$). Over this same period, we calculated daily mean wind forcing and ice drift in the SBS region, from ERA5 and Polar Pathfinder ice motion (available 1978 - 2022), respectively. We directly compared these to calculate the speed ratio of mean ice drift to mean wind in the SBS. Note that ice drift speeds from daily passive microwave measurements are lower than those measured at higher sampling rates with buoys, and both the wind and ice motion fields were averaged over the SBS. Thus, the speed ratios calculated with these products will be different than those reported earlier.

The long-term analysis is presented in Fig. 10. First, we note a strong correspondence between the number of consolidated season days with moderate ($> 5\,\mathrm{m\,s^{-1}}$) east-southeast (ESE) wind forcing over the SBS and days with large lead extents (Fig. 10e). The number of moderate ESE wind days explains 55% of the variance in open lead days, and their correlation has a slope near unity, such that each additional day of moderate ESE wind forcing in the consolidated season is likely matched by another day with open leads. Overall, lead events span a third of all consolidated season days on average over the record. However, there has been a moderate (R=0.506) increase in consolidated lead events in the SBS over the past two decades at a rate of 13 days per decade (Fig. 10b). Between 2003-2013, lead events occurred for approximately a month each consolidated season ($28 \pm 10$ days). Between 2014-2024, their average duration had increased to nearly two months ($51 \pm 14$ days).

Figure 10d shows the distributions of ice-wind speed ratios during open and compact days across the full record. Though measured with much coarser data than the SIDEx buoys in 2021, these data reinforce the finding that coastal lead events coincide with periods of enhanced ice-to-wind momentum transfer efficiency. The median speed ratio across open lead days ($1.6\,\%$) is more than twice that of compact days ($0.7\,\%$). The median across the entire record is $1\,\%$. However, the number of days with speed ratios exceeding this rate has increased moderately (R=0.466) over the 2003-2022 record, at nearly the same rate as the number of open lead days (Fig. 10c). Interestingly, despite the strong correlation between occurrences of ESE wind and lead events, the frequency of ESE wind forcing has not increased significantly over the record (Fig. 10a). This suggests that while wind forcing is a dominant control on the interannual variability of SBS lead event occurrences, other factors may be acting to increase the prevalence of lead events and enhance speed ratios over time. Mechanical weakening of a thinning sea ice

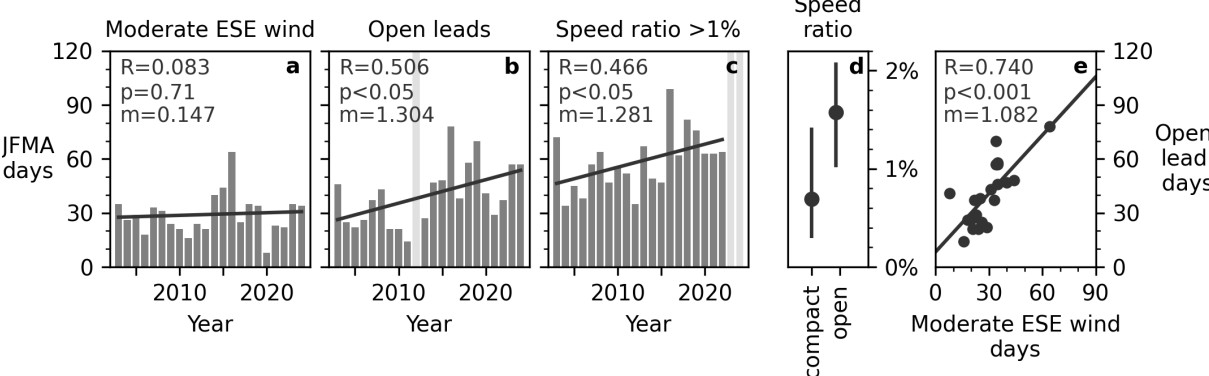

**Figure 10.** Long-term analysis of consolidated season conditions in the SBS. Number of January-April days over years 2003-2024 with (a) ESE winds (as outlined in blue in Fig. 8) exceeding $5\,\mathrm{m\,s^{-1}}$, (b) open lead extents $> 10^3\,\mathrm{km}^2$, and (c) SBS ice-wind speed ratios $> 1\,\%$. Years with missing data marked in light gray. Panel (d) shows distributions of SBS ice-wind speed ratios on open vs compact days over the full record. Points mark 50th percentiles, lines span 25th to 75th. Panel (e) plots the number of open lead days against SE wind days for each year with data. For each panel with a linear regression (line), R-values, significance (p) values and slopes (m) are listed.

cover is commonly identified as a potential driver of increased lead frequency and ice drift speeds (Rampal et al., 2009; Zhang et al., 2012; Rheinlænder et al., 2022). While 2021 saw unusually thick, old ice in the SBS and fewer lead events than most other years of its decade, its frequency of high speed ratio days was similar to other recent years which saw less expansive old ice in the SBS. Thus, how changing ice thickness may modulate the susceptibility of the ice pack to SBS lead opening events is not clear in these observations.

Rising air temperature is another factor that could explain some of the trends identified here. For instance, the strongest trends in lead occurrence take place in March, while some other months (February, April) see no significant trend in lead occurrence over the record. Simultaneously, March saw a positive trend in 2m air temperature over the record (Qu et al., 2021), and had the weakest correlation between frequency of ESE winds and lead days (R=0.529). Unseasonably warm air temperatures could reduce freezing rates substantially within leads, maintaining them for several days after an ESE wind event, reducing the correspondence between occurrences of favorable dynamic forcing and ice opening. Supporting this idea is the observation that March of 2019 saw both the warmest 2m air temperatures and the greatest prevalence of lead days over the record, despite seeing fewer than average ESE wind events. The possibility of an increasing thermodynamic contribution to spring lead maintenance could weaken the dynamic connection between wind orientation, lead occurrence, and ice drift explored in this study.

## 4 Summary and conclusions

Record winter winds in 2021 blew a large fraction of the Arctic's perennial sea ice cover into the southern Beaufort Sea (Mallett et al., 2021; Moore et al., 2022), a region typically dominated by FYI. A network of buoys was deployed throughout this region

beginning in early March 2021 as part of SIDEx. The buoys tracked MYI as it drifted west-northwestward along the Alaskan

coast, approaching the melt-prone waters of the Chukchi Sea in late April as spring breakup of the ice cover began. Ice transport throughout this period was intermittent as the winds of passing weather systems pushed the consolidated sea ice cover toward and away from coastal boundaries.

We analyzed how MYI drift across the SBS during March-April 2021 related to coastal lead opening events, during which the consolidated sea ice cover pulls away from coastal boundaries under offshore or alongshore winds. Previous studies (e.g.,

Lewis and Hutchings, 2019) have suggested that such events enhance rates of ice transport in winter, yet have typically relied on satellite-derived sea ice motion products which offer limited spatial and temporal resolution. Here we utilized hourly ice motion measurements from 23 SIDEx buoys placed throughout the SBS region, arranged in nested arrays of $15 - 100\,\mathrm{km}$ radii. These buoys tracked the cumulative transport of MYI in the SBS region over the two-month analysis period, and provided dense observations of changing sea ice drift patterns during lead opening events.

We used the $1\,\mathrm{km}$ resolution MODIS-AMSR2 sea ice concentration product to identify days when wide leads opened throughout the coastal southern Beaufort Sea. This was found to occur for just under half ($48\,\%$) of all days between 1 March and 30 April 2021, partitioned into distinct opening events which lasted 1-5 days. Opening events occurred when atmospheric highs or ridges were positioned north or east of the SBS region, driving winds from the east or southeast over the SBS region, blowing away from SBS coastlines. Most of the opening was detected along the eastern boundary of the SBS region as the SBS

ice pack slid westward along the Alaskan coast as a cohesive sheet, opening a flaw lead between Herschel and Baillie Islands. Flaw leads in the western SBS region and more localized coast-originating lead patterns were also detected. Between opening events, the SBS remained compact and consolidated with the coast. These compact periods occurred for 1-11 consecutive days, spanning $52\,\%$ of all days in the analysis. On average, these events were associated with a high pressure system positioned over the western SBS region, producing north or west winds which compressed ice against the coastal boundaries.

From 1 March to 30 April, a SIDEx buoy central to the network tracked the MYI as it drifted $365\,\mathrm{km}$ towards the west-northwest, parallel to the Alaskan coast. The vast majority of this transport ($94\,\%$) occurred on days with detected lead opening events. Just $6\,\%$ of the displacement occurred during compact days. This disparity in contributions to seasonal transport patterns is particularly striking when considering that open and compact conditions occurred for roughly equal numbers of days. To investigate the source of this disparity, we compared the ice motion to interpolated winds from ERA5 atmospheric reanalysis

for all 23 buoys throughout the analysis period. We found that median wind speeds local to the buoys on days with opening were $60\,\%$ greater than those on compact days. This moderate enhancement in wind speeds was accompanied by a more than eight-fold increase in median ice drift speeds from compact days ($1.9\,\mathrm{cm\,s^{-1}}$) to open days ($15.8\,\mathrm{cm\,s^{-1}}$).

Direct spatial comparisons of winds and ice drift suggest that the efficiency of wind-driven ice transport was enhanced on open days because the opening ice cover diverged sufficiently to reach free drift (zero ice stress) conditions as east or

southeast winds pushed the ice cover away from local coastlines. On open days, the ice most frequently drifted at $2.2\,\%$ the wind speed and $20\,\%$ to the right of the wind direction, characteristic of free drift conditions (Nansen, 1902; Leppäranta, 2011; Brunette et al., 2022). These results show that wind-to-ice momentum transfer characteristic of free drift may frequently be achieved under winter conditions when winds are oriented offshore or alongshore. On compact days, west and northwest

winds compressed the ice pack against the coast. Internal ice stresses partially or entirely balanced the wind forcing, resulting in exceptionally weak wind-to-ice momentum with ice-to-wind speed ratios most frequently between $0\% - 0.1\%$. Even under moderate wind forcing, the ice pack remained mostly stationary.

Transitions between dynamic ice regimes at the start and end of opening events was often abrupt, occurring in a matter of hours as the orientation of wind forcing over the southern Beaufort Sea shifted. At the start and end of some opening events, spatiotemporal discontinuities were measured in the sea ice velocity field around the SIDEx buoys as coastal leads directly transected the buoy network. We analyzed one of these events in detail. Thermal infrared satellite imagery confirmed that the abrupt increases in wind-to-ice momentum transfer occurred where and when leads separated the ice from the coast.

Finally, we extended our analysis of consolidated season ice dynamics in the SBS using coarser remotely sensed data to investigate conditions from January-April 2003-2024. These data showed that occurrences of moderate east-southeast winds and SBS lead events were strongly correlated, and lead events were associated with enhanced wind-to-ice momentum transfer throughout the SBS. These findings highlighted east-southeast wind events as a dominant driver of interannual variability in SBS lead occurrence during the consolidated season. However, significant increases in the prevalence of lead events and high ice-wind speed ratios over the record were not matched by trends in east-southeast wind occurrence, suggesting other factors may be acting to increase the frequency or duration of lead events over time.

The average pattern of winter and spring ice motion in the Beaufort Sea transports perennial ice from the central Arctic towards lower-latitude seas, where risk of ice melt increases. Consequently, variability in ice transport patterns throughout winter and spring has important implications for the pan-Arctic ice mass balance moving into the melt season. Recent model simulations have indicated that winters with greater lead prevalence in the Beaufort Sea see enhanced ice volume export from the region (Rheinlænder et al., 2024), and ice drift during individual coastal lead opening events can enhance transport of perennial ice across the sea (Rheinlænder et al., 2022). However, the simulations did not show a consistent relationship between winter lead prevalence and export of perennial ice from year to year.

The observational results from this study confirm the importance of ice motion associated with transient lead opening events for Beaufort Sea ice transport over seasonal time scales. In 2021, perennial ice that had moved into the southern Beaufort Sea in winter was subsequently moved westward toward the melt-prone waters of the Chukchi Sea almost entirely during coastal lead opening events occurring in spring. Given the large variability in early winter perennial ice coverage in the Beaufort Sea, it is not surprising that model simulations do not show a consistent relationship between lead activity and perennial ice export from the Beaufort Sea. In years when little perennial ice reaches the southern Beaufort Sea by early winter, opening events will mostly control advection of first-year ice. However, in years such as 2021, when a large portion of the Arctic's perennial ice cover was advected into the southern Beaufort Sea by early winter, efficient wind-driven ice drift during coastal opening events was the primary driver of the subsequent perennial ice transport. Models aiming to accurately represent changes to the pan-Arctic perennial ice distribution at seasonal timescales need to be able to represent the patterns of ice transport that occur during coastal lead opening events. These findings suggest that skillful simulation of these ice transport patterns would require accurate representations of both the atmospheric forcing that drives lead opening and the resultant changes in dynamic ice response that occurs when the consolidated ice cover opens.

*Code and data availability.* SIDEx buoy data are available at the Arctic Data Center (https://doi.org/10.18739/A2J678Z4N, Hutchings et al., 2023). 1 km SIC data (https://doi.org/10.3390/rs12193183, Ludwig et al., 2020) and 6.25 km SIC data ( https://doi.org/10.1029/2005JC003384, Spreen et al., 2008) were obtained from Universität Bremen (https://seaice.uni-bremen.de/sea-ice-concentration/). ERA atmospheric reanalysis data were obtained from the Copernicus Climate Data Store (https://doi.org/10.24381/cds.adbb2d47, Hersbach et al., 2023). Polar Pathfinder sea ice drift vectors (https://doi.org/10.5067/INAWUWO7QH7B, Tschudi et al., 2019a) and weekly sea ice age (https://doi.org/10.5067/UTAV7490FEPB, Tschudi et al., 2019b) were obtained from the National Snow and Ice Data Center. MODIS Level-1B imagery data were obtained from the NASA LAADS DAAC (Level-1 and Atmosphere Archive and Distribution System Distributed Active Archive Center, https://ladsweb.modaps.eosdis.nasa.gov/, https://doi.org/10.5067/MODIS/MYD021KM.061, MCST, 2017a, https://doi.org/10.5067/MODIS/MOD021KM.061, MCST, 2017b). VIIRS Level-1B imagery data were obtained from the NASA LAADS DAAC (https://doi.org/10.5067/VIIRS/VJ102MOD.021, VCST Team, 2021a, https://doi.org/10.5067/VIIRS/VNP02MOD.002, VCST Team, 2021b). Arctic bathymetry data were obtained from the British Oceanographic Data Centre (https://doi.org/10.5285/e0f0bb80-ab44-2739-e053-6c86abc0289c, GEBCO Bathymetric Compilation Group 2022, 2022). All python codes used in data analysis and to create figures (https://zenodo.org/doi/10.5281/zenodo.10933190, Jewell 2024) are accessible on GitHub (https://github.com/mackenziejewell/SIDEx-MYI-transport, last access: 23 October 2024).

*Video supplement.* Movie S1: Animation of Beaufort Sea ice conditions, buoy motion, and atmospheric forcing from 1 March - 30 April, 2021. Sea ice conditions shown six times daily (average 4-hour time steps) in thermal infrared MODIS imagery from the NASA/Terra satellite (MCST, 2017) and thermal infrared VIIRS imagery from the Suomi NPP and JPSS-1/NOAA20 satellites (VCST, 2021). Light shades show colder sea ice and dark shades show warmer open water. Land mask overlain in grey. Top left panel shows ERA5 sea level pressure field (Hersbach et al., 2023). 10m winds from ERA5 are overlain on imagery as grey vectors. Circles mark analyzed SIDEx buoy positions (Hutchings et al., 2023), with hourly buoy velocity vectors shown in black. Colors of buoy markers indicate ratios of buoy drift speeds to speeds of local hourly ERA5 winds.

Movie S2: Animation of Beaufort Sea ice conditions, buoy-measured ice motion, and wind-ice relations from 1 March - 30 April, 2021. Left panels show sea ice conditions six times daily from thermal infrared MODIS (MCST, 2017) and VIIRS (VCST, 2021) imagery as in Movie S1. Land mask overlain in grey. Circles mark SIDEx buoy positions (Hutchings et al., 2023), with hourly buoy velocity vectors shown in black. Colors of buoy markers indicate ratios of buoy drift speeds to speeds of local hourly ERA5 winds (Hersbach et al., 2023). From top to bottom, panels on right show the cumulative transport of buoys along the west-northwest (WNW) direction, the wind-ice deflection angle between buoys and local winds (clockwise positive), the speed ratio of buoys to local winds, and the total Southern Beaufort Sea extent with detected openings.

*Author contributions.* MJ and JH participated in the conceptualization of the research plan. MJ developed the methodology, carried out the analysis, and prepared the paper. MJ, JH, and AB interpreted and discussed the results and contributed to the review and editing of the paper.

*Competing interests.* The contact author has declared that none of the authors has any competing interests.

*Acknowledgements.* This research was supported by the Office of Naval Research (grant no. N000141912604). MJ was partially supported by a NASA FINESST grant (no. 80NSSC21K1601), and AB was partially supported by the NASA Internal Scientist Funding Model in the Cryospheric Sciences Program. We would like to thank the editor and two anonymous reviewers for their comments which improved this manuscript.

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
