# Peer review of "Spring 2021 sea ice transport in the southern Beaufort Sea occurred during coastal lead opening events"

_EGUsphere, 2024_

## Author Comment (AC1)

**Response to Referee #1 Comments**

We would like to thank Referee #1 for evaluating our manuscript and for providing constructive feedback. We are currently working on adjusting some of the language in the paper and adapting figures as recommended by the reviewer and respond to specific comments below. We show referee comments in gray text and our response in blue text.
* * *
**Referee #1 Comment** on "Spring 2021 sea ice transport in the southern Beaufort Sea occurred during coastal ice opening events" by Jewell et al., EGUsphere [preprint], https://doi.org/10.5194/egusphere-2024-1097

**Citation**: https://doi.org/10.5194/egusphere-2024-1097-RC1

**Summary**

The presented paper evaluates how coastal ice opening events in the sea ice of the southern Beaufort Sea (SBS) in spring 2021 modulate the efficiency of wind forcing on sea-ice drift and the associated transport of multiyear sea ice towards the Chukchi Sea. It is hypothesized, that coastal openings favour the free drift of sea ice in the SBS and thereby potentially contribute to the observed annual losses of MYI. The authors use remote sensing data (combined MODIS/AMSR2) to identify sea-ice opening events, i.e. flaw leads, a SIDEx buoy array as well as Polar Pathfinder sea-ice drift data to track the transport of ice and ERA5 reanalysis data to derive associated atmospheric conditions. The goal of the presented research is to relate sea-ice drift in the SBS to coastal ice opening events and identify potential implications for Arctic multi-year ice loss.

**General comments and decision**

The paper presents a well-written study with a meaningful structure that is nice to follow by the reader. The presentation of the results is well thought out and the discussion is placed in good context with current research. Also, the supplements provide a valuable add-on to the presented results. However, I think the paper in its present form needs to be strengthened in pointing out what new lessons were learned and in how far these are important for moving our understanding forward in this field. It is a well-documented and detailed case study of sea-ice dynamics in the Beaufort Sea in spring 2021, which finally misses to embed its findings into a meaningful longer-term context. Although the introduction and discussion aim to connect to this context, the limitation to derive coastal openings and associated drift patterns for a period of 2 months only represents a weakness of this paper (how special was 2021 in terms of opening events?). This work would gain

strength and meaning if at least ice opening events were analysed on longer time scales (trends?) and then compared to what was happening in 2021.

We appreciate the reviewer's comments on the strengths of this case analysis of 2021 sea ice transport, and suggestions to contextualize the findings from 2021. We agree this could strengthen the conclusions of the manuscript, and are working on a figure to show this by comparing across winters 2018-2024 (the period over which the high-resolution SIC product is available). NSIDC sea ice motion is only available through 2022 at daily resolution, so we compare ice motion 2018-2022.

See the figure below, which shows the number of open days in March - April from 2018 - 2024 (panel a), and which compares distributions of SBS ice drift speeds on open and compact days (panel b, points marking 50th percentile, lines spanning 25th-75th percentiles). 2021 saw the fewest days in March-April with large lead area compared to the three years prior and after. For each year, open days generally saw faster SBS drift speeds than compact days (blue vs. orange in panel b), though more opening events in a given year was not necessarily associated with faster average seasonal ice drift speeds. Overall 2021 ice drift speeds (black distribution) fell in the typical range of values over the 1980-2022 record (shown in gray above panel b). This is a useful figure which we could include in the manuscript for context. However, we feel that extending the analysis much beyond this, to include trends analyses for example, might stray too far from the primary focus of the manuscript, which is a detailed analysis of 2021 ice dynamics.

[Figure]

Finally, the final paragraph of the manuscript speaks to the new lessons learned (nearly all the ice transport in this year is partitioned into these rapid drift events lasting just half the time), and how they move our understanding forward in this field (answering open questions from modeling studies about how patterns of motion during lead events might impact ice transport).

Another major concern I have is that the paper postulates a causal relationship between ice opening events and free ice drift in the SBS, which I think is a misleading concept.  I think

that the factor that drives both, flaw lead formation and free ice drift, is wind direction and wind speed in combination with coastal geometry, which has been shown also in other papers investigating flaw lead and polynya dynamics. While this relationship is documented and mentioned in the submitted paper, the free ice drift events are finally explained by the occurrence of sea-ice opening events, which I think is a bit confusing. In this context I also think that the title of the paper should be adjusted.

We also greatly appreciate the reviewer's comments about language in the manuscript implying that leads cause ice motion or free drift. This was not our intention, and we plan to look for and adjust text which could be interpreted this way. In particular, we are currently working on restructuring some of the text at the end of the introduction, and other phrases throughout the manuscript which may create confusion around this point.

Our aim was to highlight that the formation of coastal leads signals where and when spatiotemporal discontinuities in the ice drift field are occurring. As shown in the case study of event E5 (Figure 9), abrupt transitions from stationary to free drift-like motion (and back again) occurs across the coastal leads in 2021. The leads form *because* portions of the ice cover start moving, which can occur once winds are oriented favorably relative to coastal boundaries.

Finally, it was suggested that we change the title so as not to imply that leads cause free drift. In its current state, the title "Spring 2021 sea ice transport in the SBS occurred during coastal ice opening events" does not mention free drift and was specifically worded to avoid an implication of causality. It qualitatively summarizes the primary quantitative finding of this study: 94% of the alongshore ice transport during this period occurred during time when coastal leads were open. We do plan to change "ice opening events" to "lead opening events" as per the suggestion below.

**Detailed comments**

The terms "lead opening events" and "ice opening events" are both used to describe the same process. Please consider using only one.

Thank you for this suggestion, we will replace "ice opening events" with "lead opening events" throughout the text.

Page 4, bottom: I don't think it needs this equation if the approach is described in the text.

We will remove this equation from the text.

L103 ff: ["Physically, this …"]. I think this statement needs a better explanation or reference (other than WMO) and that the value of 200m is highly sensitive to the amount of thin ice in the leads.

Here, WMO was cited to describe that a lead with 200m width falls under the WMO lead size classification as a "medium fracture." We will mention that this product sometimes sees small negative SIC biases over frozen ice within leads, but this is acceptable for our purposes given that we are aiming to detect recent ice openings even if they are freezing over.

L118: "SIC-derived area with opening…" Not clear, what exactly is meant.

We will rephrase as "calculated open area"

Figure 5: Can buoy positions be shown in 5k?

We had considered showing this, but given that 5k comprises many individual events, including all the tracks looks quite messy. See figure below. We decided against showing the figure with buoy tracks in all panels for visual clarity.

[Figure]

L274: What was the average wind direction for compact and open days?

These wind directions are described in Lines 298-299: opening events are most frequently associated with east winds, while compact periods are associated with west winds. The distributions of wind directions for open and compact days are also shown in Figure 7 and summarized in Figure 8.

Figure 6 is quite small. Can it be widened?

We can widen this in the revised version as below.

[Figure]

Figure 9 Please show where this regional subset (c-f) is located in the study area (Fig. 2).

Thank you for this suggestion, we plan to expand the mapped region as shown below to include the entire SBS which should make the map extent more clear, and also provide more information about the conditions across the entire region.

[Figure]

L 413: "…usually remained quite high…" Can that be shown here?

This is visible in the SIC maps of Figure 5, but as it is not a key point of the analysis, we plan to remove this sentence altogether.

---

## Author Comment (AC2)

**Response to Referee #2 Comments**

We would like to thank Referee #2 for evaluating our manuscript and for providing constructive feedback. We are currently working on adjusting some of the language in the paper as recommended by the reviewer and respond to specific comments below. We show referee comments in gray text and provide our response in blue text.
* * *
**Referee #2 Comment** on "Spring 2021 sea ice transport in the southern Beaufort Sea occurred during coastal ice opening events" by Jewell et al., EGUsphere [preprint], https://doi.org/10.5194/egusphere-2024-1097

**Citation**: https://doi.org/10.5194/egusphere-2024-1097-RC2

**Summary:**

This paper analyzes the drift of a tight array of GPS buoys in the Southern Beaufort Sea in response to wind speed and direction. This paper is generally well written, and I recommend acceptance after very minor revisions. Most of my comments are suggestions on grammar and style.

**Details:**

1)    Lines 12 and 15: These sentences imply that lead opening causes increases SBS ice motion, but I would argue that both the lead opening and increased ice motion are both a consequence of favorable winds away from the coast. I suggest rewriting this as "These results quantify the disproportionate contribution in offshore winds towards breaking the sea ice away from the coast, lead opening and SBD ice transport during Spring 2021, …". While this idea may not be new, this paper discusses these processes in wonderful detail that warrants publication of these results.

We thank the reviewer for this feedback. Noting that another reviewer also pointed this out, we understand this reads as "lead opening causing sea ice motion." As we do not want to suggest this is the case, we will incorporate the suggested re-phrasing into our revised manuscript.

2)    Line 48: suggest rewriting this as "The consolidated season in the Beaufort Sea extended through March and April 2021..." Since the subject here is the "consolidated season", rather than March and April.

3)    Line 50: Delete "During the consolidated season, ". Too wordy.

4) Line 51: Change "Of sufficient magnitude, these internal stresses can", to "The internal stresses can be of sufficient magnitude to cause the ice pack..."

We thank the reviewer for grammatical suggestions 2-4, and will shorten and re-arrange these phrases accordingly.

5) Lines 57-58: Rather than "that lose and remain", I think the authors mean "that is loose and remains".

While we did originally mean "that lose and remain," we realize this sentence is redundant with more clear explanations that follow, so will remove it in the revised manuscript.

6) Tschudi et al. 2019b paper should be cited in Figure 1 caption or in the Data section.

We thank the review for this suggestion. While we cited the data product, we had not cited the associated paper (https://doi.org/10.5194/tc-14-1519-2020) and will make sure to include this in the revised manuscript. We will add this to the body of the text where the sea ice age record is first referenced.

7) The first paragraphs of new sections aren't indented. Is this the preferred style of the journal?

We are using the template provided by the journal, though formatting does change between the initial submissions and final published manuscripts.

8) Line 246: Suggest change "just" to "only".

We will make this adjustment in the revised manuscript.

9) Lines 296 and 301. I suggest West-Northwesterly rather than repeat the "erly" suffix.

We appreciate this suggestion to shorten the named wind directions and will edit accordingly.

10) The supplemental movies are wonderful! Lots of insight/information buried in these beyond what is discussed in the paper. I can see these being used/cited in many papers/presentations on sea ice.

We thank the reviewer for this reflection, and were excited to share them so they could also be used by others who will have additional insights beyond our own about all the information these observations provide.